ecology

*Megachile rotundata*, soil moisture, neonicotinoids, imidacloprid, nectar, *Phacelia tanacetifolia*

**Author for correspondence:**
Jacob M. Cecala
e-mail: jmcecala@gmail.com

# Pollinators and plant nurseries: how irrigation and pesticide treatment of native ornamental plants impact solitary bees

Jacob M. Cecala and Erin E. Wilson Rankin

Department of Entomology, University of California, 900 University Ave, Riverside, CA 92521, USA

JMC, 0000-0002-6224-8517; EEWR, 0000-0001-7741-113X

A key conservation goal in agroecosystems is to understand how management practices may affect beneficial species, such as pollinators. Currently, broad gaps exist in our knowledge as to how horticultural management practices, such as irrigation level, might influence bee reproduction, particularly for solitary bees. Despite the extensive use of ornamental plants by bees, especially little is known about how irrigation level may interact with insecticides, like water-soluble neonicotinoids, to influence floral rewards and bee reproduction. We designed a two-factor field cage experiment in which we reared *Megachile rotundata* (Fabricius) (Hymenoptera: Megachilidae) on containerized ornamental plants grown under two different irrigation levels and imidacloprid treatments (30% label rate dosage of a nursery formulation or an untreated control). Lower irrigation was associated with modest decreases in nectar volume and floral abundance in untreated plants, whereas irrigation did not affect plants treated with imidacloprid. Furthermore, higher irrigation decreased the amount of imidacloprid entering nectar. Imidacloprid application strongly reduced bee foraging activity and reproduction, and higher irrigation did not offset any negative effects on bees. Our study emphasizes the impact of a nursery neonicotinoid formulation on solitary bee foraging and reproduction, while highlighting interactions between irrigation level and neonicotinoid application in containerized plants themselves.

## 1. Introduction

Ornamental plant nurseries represent a major agricultural sector that remains relatively unexplored with regard to its support of local insect communities. While ornamental plants and the urban greenspaces they occupy are well known to serve as foraging resources for pollinators [1,2], the role of horticultural nurseries as bee foraging habitat has just recently received attention [3–6]. While these facilities occupy less land area than conventional row crops do [7], their high floral diversity [3–5] and the potential for exposure to elevated concentrations of insecticides in floral resources [8–10] render nurseries pertinent to study from the perspective of wild bee ecology. However, we lack quantitative knowledge on how local management practices in horticulture may interact to affect resident wild bees, particularly solitary species. Relevant in-field management practices for bees in any agricultural area include those affecting vegetation quality, agrochemical input, and soil characteristics [11]. Interactions between local management practices, such as whether one mitigates or exacerbates the effects of another, are only recently being investigated using manipulative experiments [12,13].

Pesticide use is a management practice influencing wild bee conservation in all agroecosystems. Throughout the past decade, concerns have arisen over systemic insecticide use in ornamental plants [8,9,14]. Neonicotinoid insecticides, for example, tend to be applied at higher levels in ornamental crops than in food crops [10,14] due to differences in pest management goals, formulations,

and application methods. Multiple factors influence the amount of these systemic insecticides entering pollen and nectar of ornamental plants [15,16], but we have a poor understanding of these factors and how they interact [17]. Recent studies revealed that increased local floral resource availability may buffer against some negative impacts of pesticide exposure in solitary bee populations [12,13]. However, supplementation of floral resources [11] for bees is unlikely to be financially incentivizing in nurseries, as most ornamental crops rarely require pollination services. Whether such a buffering effect would be observed in the context of higher pesticide concentrations potentially found in floral resources of ornamental plants remains unresolved [16].

Irrigation is another horticultural management practice, which is also tied to water conservation, pest management, and environmental runoff [18], that can potentially impact bees. Reduced water availability may negatively affect floral traits such as the quantity and quality of floral rewards [19–22], and may cause cascading negative effects on pollinator visitation [23,24] and crop yield [25,26]. However, few experimental studies directly link water availability to pollinator fitness. Wilson Rankin et al. [22] found that reduced water yielded lower quality pollen and nectar, which negatively impacted the fitness of eusocial bees that were fed nutritionally equivalent artificial diets. We know of no studies to date that investigate how plant irrigation level influences solitary bee reproduction and fitness under field conditions.

Whether the benefits to bees from increased irrigation offset the detrimental effects of pesticide exposure remains unknown. Potential interactions between irrigation level and pesticide use both on bees and floral resources have received little attention. Understanding these interactions is critical, however, for systemic water-soluble insecticides like neonicotinoids. Water availability can influence the rates of several physical and biological processes that could potentially affect the amount of a neonicotinoid present in floral resources [17]. These processes include, among others, plant transpiration [27,28], leaching of insecticide ingredients from soil [29], and production of flowers and floral resources by the plant [19,22]. Moreover, nursery plants are confined in containers, unlike plants in most other agricultural areas. Containerization limits a plant's root zone, which could influence how water affects uptake and leaching of neonicotinoids relative to plants grown in the ground. Addressing such potential interactions between irrigation and systemic pesticide use on bee fitness could specifically benefit nurseries by improving consumer perceptions of their pollinator stewardship efforts [8].

To address these knowledge gaps, we reared alfalfa leafcutting bees ('ALCB', Megachile rotundata) on containerized ornamental plants from a nursery grown under different irrigation and neonicotinoid application regimes. The ALCB is a solitary, cavity-nesting species managed as a pollinator in North America, emerging as a model for studying solitary bee biology [30] and pesticide risk assessments [31]. We test the hypothesis that the irrigation level of potted ornamental plants can modulate the effects of systemic pesticide application on bees. We predict that plants receiving higher irrigation levels will provide more abundant or higher quality resources for bees, which will offset some negative influences of neonicotinoid exposure on foraging and reproduction. Specifically, we expect that bees reared on higher irrigated plants will exhibit higher foraging activity and reproduction relative to those reared on plants irrigated at a lower level, both for bees

reared on untreated and neonicotinoid-treated plants. Through manipulative experiments, we aim to improve our understanding of how local management practices interact to affect the population stability of this ecologically important pollinator.

## 2. Methods

We examined the effects of imidacloprid application and irrigation level on floral nectar and ALCB reproduction using a fully crossed randomized block design. From March 2018 to July 2020, we maintained 20 field cages (each 5.8 m$^3$) in a 0.30 ha plot at University of California Riverside Agricultural Operations (33.965° N, 117.341° W). We used four cages for the nectar experiment and 16 for the ALCB reproduction experiment. Each cage served as a replicate mesocosm simulating conditions at a containerized nursery.

### (a) Nectar experiment

We began growing lacy phacelia (Boraginaceae: Phacelia tanacetifolia Benth.) from seed in UC Soil Mix III (agops.ucr.edu/soil-mixing) in 2 l pots in a greenhouse on 29 October 2019. We focused on containerized plants, as soil dynamics for plants in containers likely differ from those experienced by plants growing in the ground. We selected phacelia due to its attractiveness to bees, including ALCBs [31], and its abundant floret production [32]. We assigned 180 plants to one of six treatments resulting from the crossing of irrigation level and imidacloprid treatment (table 1), organized into two experimental blocks of two cages each (electronic supplementary material, figure S1a), resulting in 45 plants per cage.

Plants were moved into cages on 14 January 2020, and we inserted an individual high- or low-flow irrigation spike (Primerus Products, Encinitas, CA, USA) into each pot. These spikes are widely used in nurseries for container irrigation, and we selected spikes representing the lowest and highest flow rates for 2 l pots. High-flow spikes emitted 2.6 times more water as low-flow spikes (electronic supplementary material, figure S2), resulting in 23% higher average midday volumetric water content (VWC; electronic supplementary material, figure S3a). All pots were automatically irrigated simultaneously over the soil surface once per day at 06.00 h for 60 s, increased to 120 s whenever any plants visibly wilted due to heat. The ground inside each cage was lined with a fabric barrier to prevent plants from rooting outside pots.

For imidacloprid treatments, we applied Marathon® 1% Granular (OHP, Bluffton, SC, USA), a commercial nursery formulation, to pots on 28 January 2020. Formulated for use in greenhouses and nurseries, Marathon® consists of 1% imidacloprid and 99% inert ingredients by mass. Granular formulations allow the active ingredient to leach more slowly from potting soil than liquid drenches [33]. The label rate dosage for a 2 l pot equates to 1.4–2.0 g formulation. We used approximately 30% the label rate as our 'high' dosage to be conservative, as near total ALCB mortality occurred when separate plants were treated at label rate (JM Cecala 2018, unpublished data). In this experiment, we also included a 'low' dosage treatment of approximately 3% the label rate (table 1).

Phacelia bloom began in early May 2020 and lasted six weeks. On each of six separate days during the bloom period, we measured floral nectar volume between 10.30 and 13.30 h (daily high temperatures between 22.2 and 36.1°C) in randomly selected flowers in each treatment using microcapillary tubes (Drummond Scientific, Broomall, PA, USA) and handheld calipers ($N = 13$ flowers × 4 cages × 6 days = 312 flowers). We also quantified total sugar concentration of samples on three of these days using a refractometer (Eclipse, Bellingham + Stanley, Tunbridge Wells, Kent, UK) ($N = 7$–8 flowers × 4 cages × 3 days = 94 flowers). On each of eight separate days during bloom, we collected nectar during the same time of day, pooled samples within treatments

**Table 1.** Treatments and sample sizes, as numbers of plants, resulting from the crossing of irrigation and imidacloprid treatments in the phacelia (*P*) nectar and solitary bee (*B*) reproduction experiments. In the latter, the low imidacloprid dosage treatment was excluded. Also noted is the imidacloprid mass added to each pot, where 0.1 g Marathon® = 1 mg imidacloprid, and the proportional label rate (LR).

| | | | irrigation | |
| --- | --- | --- | --- | --- |
| | | | low | high |
| imidacloprid | **control** | | *P*: 30 | *P*: 30 |
| | 0 mg imidacloprid = 0% LR | | *B*: 120 × 2 yr | *B*: 120 × 2 yr |
| | **low dosage** | | *P*: 30 | *P*: 30 |
| | 0.5 mg imidacloprid = 3% LR | | *B*: excluded | *B*: excluded |
| | **high dosage** | | *P*: 30 | *P*: 30 |
| | 5.0 mg imidacloprid = 30% LR | | *B*: 120 × 2 yr | *B*: 120 × 2 yr |

within cages ($N = 3$ pooled samples × 8 days = 24 samples (four per treatment)), and quantified imidacloprid residues in nectar via an enzyme-linked immunosorbent assay (ELISA). We used a QuantiPlate™ kit (EnviroLogix, Portland, ME, USA) and microplate spectrophotometer (Thermo Fisher, Waltham, MA, USA), which provide similar quality data to high-performance liquid chromatography–tandem mass spectrometry (HPLC/MS-MS) at less cost [16]. Imidacloprid metabolites, also toxic to bees [29], cross-react in the assay. Thus, assay results reflect the total concentration of the parent compound and its metabolites. Samples were diluted 10- to 100-fold before analysis as needed to complement the kit's quantification range.

## (b) ALCB reproduction experiment

In the other 16 field cages, we placed the California-native ornamental plants *Erigeron glaucus*, *Sphaeralcea ambigua*, and *Baileya multiradiata* in 21 pots (not previously treated with insecticides), purchased from a local native plant nursery (Valley Center, CA, USA) in 2019 and 2020. We selected these species based on their popularity at nurseries, drought tolerance to ensure bloom in low irrigation conditions, and from surveys of wild bee visitation at nurseries [34]. We also included pots of lacy phacelia and alfalfa (*Medicago sativa* L.). The ratio of plant species in each cage varied slightly between years due to availability (electronic supplementary material, table S1). We divided field cages into four experimental blocks, with one cage assigned to each of four treatments (table 1; electronic supplementary material, figure S1b). The low dosage (approx. 3% label rate) imidacloprid treatment was not included in the bee reproduction experiment.

We manipulated plant irrigation levels as in the phacelia nectar experiment. Although soil VWC varied across plant species, VWC in high irrigation pots was 44% higher than that of soil in low irrigation pots (electronic supplementary material, figure S3b). In imidacloprid treatment cages, we applied Marathon® four weeks before introducing bees to allow for translocation (as in [14]). We applied Marathon® to each plant species *except* alfalfa, which we anticipated would serve as the principal leaf clipping source for nesting female ALCBs, to ensure bees were primarily exposed to imidacloprid through consumption of pollen and nectar, and not via leaf tissue clipping.

In each cage, we provided one nest block constructed according to United States Department of Agriculture-Agricultural Research Service (USDA-ARS) specifications [35] facing southeast. Each block contained 60 drilled tunnels into which we inserted paper straws (diameter 5 mm, length 12.7 cm). We purchased commercially reared ALCB pre-pupae from Canada (JWM Leafcutters, Parkside, SK, Canada) and allowed them to develop into adults in an incubator at 30.3 ± 0.1°C, 57.2 ± 0.4% relative humidity (mean ± s.e.) and a 12 h light–dark cycle.

Emergence occurred after 21 days. In mid-June 2019 and 2020, we introduced 30 male and 20 female ALCBs inside each cage to approximate sex ratios in commercial populations [36].

Over the following six weeks, two to three times per week, we recorded floral abundance (for each plant species and the entire cage) and ALCB foraging activity in each cage using ordinal indices. For floral abundance, we assigned: '0' if no flowers were present, '1' if a few flowers were present, '2' if flowers covered 10–50% the cage area, and '3' if flowers covered greater than 50% the cage area. For bee foraging activity, we visually monitored the inside of each cage for 10 s (similar to [32]) and assigned: '0' if no foraging bees were visible, '1' if 1–3 bees were visible, '2' if 4–10 bees were visible, and '3' if greater than 10 bees were visible. We also recorded ambient temperature during observations.

After six weeks, we collected all straws and labelled and weighed each straw individually. After three weeks of storage at 22°C, straws were kept at 5°C over the winter. After at least four months, straws were incubated again. However, no bees emerged in either year, potentially due to the lack of a fluctuating temperature regime during cold storage [37], insufficient quantity or quality of pollen provisions, or adults not entering diapause (though we did not observe any non-diapausing adults emerging) [38]. To assess reproduction, we dissected straws and inspected the contents, quantifying incomplete cells (leaf pieces not fully formed into a cell), empty cells (fully constructed, but with no contents), cells containing pollen provisions but no brood, and cells with brood (and the developmental stage).

## (c) Statistical analysis

We conducted all analyses in R (v. 3.3.3) [39]. All means are reported ± s.e. In all models, we checked for collinearity using function 'vif' (*car*) [40]. To assess how treatments impacted volume, sugar concentration, and imidacloprid concentration of phacelia nectar, we constructed linear mixed models (LMMs) using function 'lmer' (*lme4*) [41]. We included as fixed effects irrigation level and imidacloprid treatment (and their interaction) and number of days elapsed since imidacloprid application. In the volume and sugar concentration models only, we included daily high temperature, known to influence nectar secretion in phacelia [32]. Cage nested within block served as random effects. Volume and imidacloprid concentration were $\log_{10}(x + 1)$-transformed. In all models, we used function 'emmeans' (*emmeans*) [42] for post-hoc comparisons (Tukey's honestly significant difference (HSD) tests) as appropriate.

To assess treatment effects on indices of cage-level floral abundance and ALCB foraging activity, we constructed additional LMMs. We included as fixed effects irrigation level, imidacloprid treatment, and year (and all interactions), and number of days elapsed since bees were added to cages. For

the bee foraging activity model, we also included ambient temperature during the observation. We again included cage nested within block as random effects. To assess how treatments influenced nest initiation by ALCBs, we constructed a generalized linear mixed model (GLMM) using function 'glmer' (lme4) and a logit link. In this model, we treated each straw (empty or not) as a replicate, noting whether there was *any* evidence of nest construction or not. We included the same fixed and random effects as the bee foraging activity model. Furthermore, to determine if treatments influenced (per cage) the number of cells (incomplete or complete) constructed, number of cells *containing brood*, or *proportion* of cells containing brood, we constructed LMMs with the aforementioned predictor variables. We square-root transformed the number of cells per cage and number of cells containing brood per cage. Finally, to determine if female ALCBs clipped plants *other* than alfalfa (which would further expose them to imidacloprid), we used Fisher's exact tests. We tested if the number of cells (incomplete or complete) constructed with versus without alfalfa (never treated with imidacloprid), differed between imidacloprid or irrigation treatments.

## 3. Results

### (a) Nectar experiment

Nectar volume was 19% higher in phacelia plants receiving high irrigation versus low irrigation (figure 1*a*; $F_{1,9} = 9.68$, $p = 0.0127$). Despite a significant effect of imidacloprid dosage, nectar volume differed by only 2.5% among imidacloprid treatments ($F_{2,302} = 3.38$, $p = 0.0352$). The effect of irrigation was mainly driven by an interaction with imidacloprid dosage (figure 1*a*; $F_{2,302} = 6.30$, $p = 0.00210$), specifically by plants in the control (54% higher volume in the high irrigation treatment) and low dosage (43% higher volume) treatments. In contrast, there was no irrigation effect on nectar volume for plants treated with the high imidacloprid dosage. Nectar volume declined with higher daily high temperatures ($F_{1,302} = 45.51$, $p < 0.0001$) and increasing time since imidacloprid application ($F_{1,302} = 80.37$, $p < 0.0001$).

Nectar sugar concentration was not correlated with any of our metrics. It was unaffected by irrigation (figure 1*b*; $F_{1,9} = 0.64$, $p = 0.443$) and imidacloprid dosage ($F_{2,84} = 3.02$, $p = 0.0540$; interaction: $F_{2,84} = 0.0819$, $p = 0.921$), although nectar from plants treated with a high dosage had 20% higher sugar concentration than nectar from the control treatment. Nectar sugar concentration did not vary with days since pesticide application ($F_{1,84} = 0.148$, $p = 0.702$), daily high temperature ($F_{1,84} = 0.0685$, $p = 0.794$), or nectar volume per flower ($F_{1,92} = 1.51$, $p = 0.223$).

As expected, imidacloprid dosage was positively correlated with imidacloprid concentrations in phacelia nectar (figure 1*c*; $F_{2,15} = 16.9$, $p = 0.000140$). Nectar from flowers in the high dosage contained the highest concentrations at $55 \pm 22$ ppb, while nectar from flowers in the low dosage contained $7.3 \pm 1.9$ ppb and did not differ in imidacloprid concentration from control nectar ($3.2 \pm 0.8$ ppb; however, control nectar samples fell below the ELISA's lower quantification limit). There was no effect of irrigation alone on imidacloprid concentration in phacelia nectar ($F_{1,5} = 0.434$, $p = 0.541$). We detected an interaction between irrigation and imidacloprid (figure 1*c*; $F_{2,15} = 5.28$, $p = 0.0183$) such that in the high dosage treatment, imidacloprid concentration in nectar from low irrigation plants was 3.5 times higher than in nectar from high irrigation plants. Imidacloprid

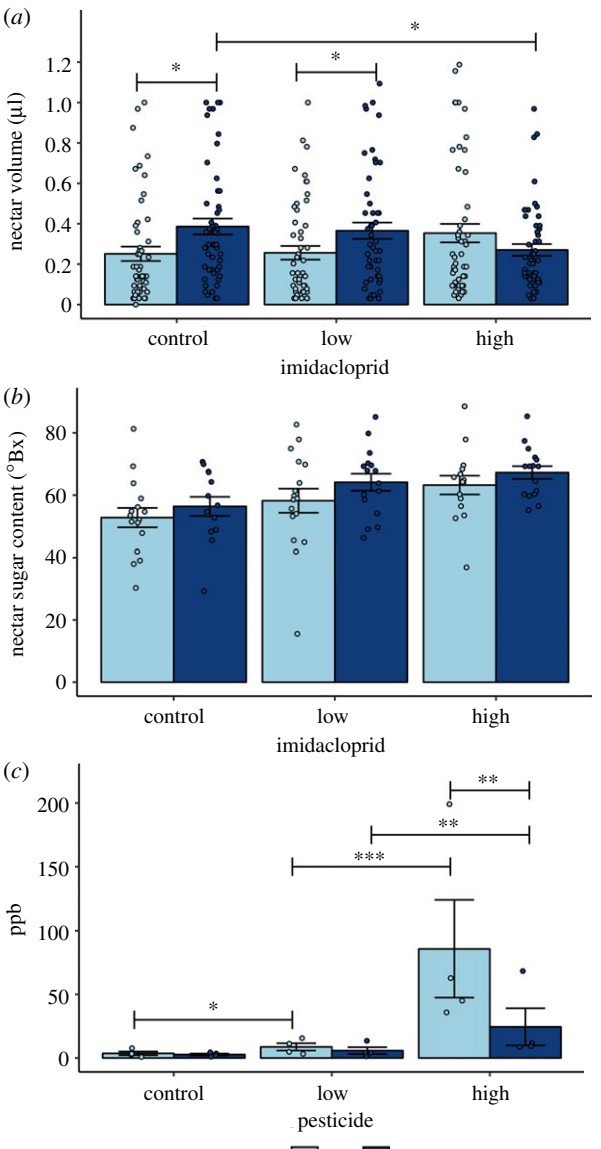

**Figure 1.** Phacelia floral nectar characteristics in response to irrigation and imidacloprid treatments: (*a*) nectar volume, (*b*) nectar sugar concentration, and (*c*) imidacloprid concentration. Points represent raw data, while bars and whiskers show mean ± s.e. Lines indicate significant comparisons from post-hoc Tukey's HSD tests (*0.01 < *p* < 0.05; **0.001 < *p* < 0.01; ***p* < 0.001). (Online version in colour.)

concentration declined with days elapsed since application (slope: −0.0298; $F_{1,8} = 14.4$, $p = 0.00527$) during our sampling period (14.6−18.6 weeks post-application). Excluding measurements from untreated control plants from this model yielded a similar result (slope: −0.0334; $F_{1,7} = 13.4$, $p = 0.00854$).

### (b) ALCB foraging and reproduction experiment

We did not detect a significant effect of irrigation ($F_{1,60} = 0.379$, $p = 0.541$) or imidacloprid ($F_{1,60} = 1.51$, $p = 0.223$) on cage-level (across all plant species) floral abundance, though there was substantial variation across individual plant species in how floral abundance responded to our treatments (electronic supplementary material, figure S4). However, there was an interaction between irrigation and imidacloprid (figure 2*a*; $F_{1,60} = 6.82$, $p = 0.0114$) such that, in control cages, cage-level floral abundance was higher in high irrigation cages (index: $1.91 \pm 0.07$) than in low irrigation cages ($1.56 \pm 0.06$). For

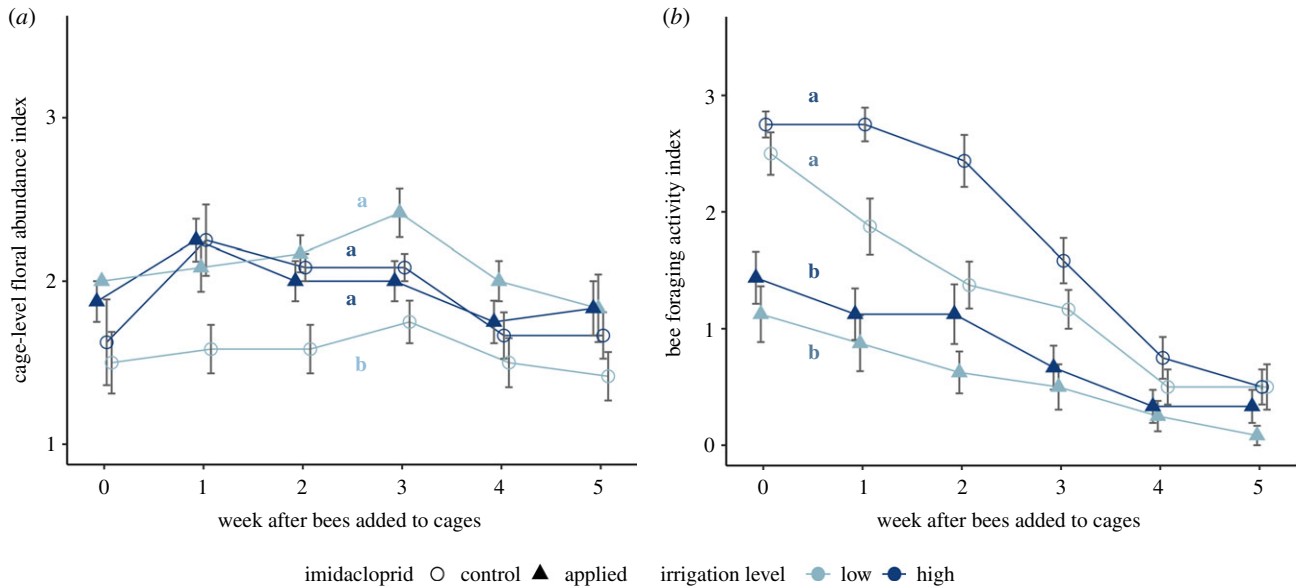

**Figure 2.** Irrigation and imidacloprid effects on (*a*) cage-level floral abundance and (*b*) ALCB foraging activity, both years combined. Note data are binned by week, with week '0' representing all data collected within the first week after bees were added to cages. Within each week, points are jittered horizontally to improve visibility. Points and whiskers represent mean ± s.e. Lines not connected by the same lowercase letter were significantly different ($p < 0.05$) in post-hoc Tukey's HSD tests. (Online version in colour.)

cages in the imidacloprid treatment, floral abundance did not differ between high irrigation (index: $1.96 \pm 0.06$) and low irrigation ($2.09 \pm 0.06$) treatments, nor did cages in the imidacloprid treatment differ from cages in the control, high irrigation treatment. These patterns were consistent between study years (imidacloprid × irrigation × year interaction $F_{1,251} = 0.394$, $p = 0.531$), and cage-level floral abundance did not differ between years ($F_{1,251} = 0.777$, $p = 0.379$). Floral abundance declined slightly over time ($F_{1,251} = 5.00$, $p = 0.0263$) while bees were in cages.

ALCBs were observed foraging on all five plant species in cages. ALCB foraging activity declined steeply over time ($F_{1,314} = 246.5$, $p < 0.0001$) and was lower in imidacloprid-treated cages than in control cages (figure 2*b*; $F_{1,14} = 24.7$, $p = 0.000224$). Bee foraging activity was slightly lower in 2020 ($F_{1,314} = 7.05$, $p = 0.00835$), and the negative effect of imidacloprid treatment was more pronounced in 2019 (imidacloprid × year interaction $F_{1,314} = 9.14$, $p = 0.00271$). Foraging activity was unaffected by irrigation treatment (figure 2*b*; $F_{1,13} = 1.94$, $p = 0.186$, irrigation × imidacloprid interaction $F_{1,13} = 0.242$, $p = 0.630$), and ambient temperature during observation periods ($F_{1,314} = 1.09$, $p = 0.297$).

ALCBs in imidacloprid-treated cages initiated on average only 4% the number of nests as in control cages, irrespective of nest contents (figure 3*a*; $\chi^2_1 = 6.93$, $p = 0.00849$). We observed no difference in nest initiation between irrigation treatments ($\chi^2_1 = 0.0254$, $p = 0.873$), and no interaction with imidacloprid treatment ($\chi^2_1 = 0.0226$, $p = 0.881$). Nest initiation in cages was not correlated with higher mean bee foraging activity ($\chi^2_1 = 3.52$, $p = 0.0606$) or mean cage-level floral abundance ($\chi^2_1 = 1.01$, $p = 0.315$), and did not differ between years ($\chi^2_1 = 0.340$, $p = 0.560$).

In control cages, nest mass ($0.143 \pm 0.009$ g, $n = 94$) was unaffected by irrigation treatment ($F_{1,5} = 0.228$, $p = 0.653$), study year ($F_{1,63} = 0.215$, $p = 0.645$), mean floral abundance ($F_{1,14} = 2.41$, $p = 0.143$), or mean ALCB foraging activity ($F_{1,29} = 1.93$, $p = 0.176$). Only four nests (mass: $0.090 \pm 0.021$ g) were initiated across all imidacloprid-treated cages,

so we could not statistically test for the effect of imidacloprid on nest mass. Mean nest mass per cage did not decline with the number of nests constructed per cage ($F_{1,5} = 0.432$, $p = 0.540$), suggesting floral and nesting resources within cages were not limiting for bees.

ALCBs in the imidacloprid treatment constructed only 5.3% as many total cells (including incomplete cells and cells without brood) per cage as bees in the control treatment (figure 3*b*; $F_{1,21} = 13.31$, $p = 0.00154$), and constructed only 5.8% as many cells *containing brood*—as opposed to being empty or containing only pollen provisions—as bees in the control treatment (figure 3*b*; $F_{1,21} = 5.25$, $p = 0.0325$). However, the proportion of cells per cage that contained brood ($0.14 \pm 0.03$, $n = 32$) did not vary consistently with any treatment (figure 3*b*; imidacloprid: $F_{1,20} = 0.462$, $p = 0.504$; irrigation: $F_{1,20} = 1.02$, $p = 0.323$; interaction: $F_{1,20} = 1.79$, $p = 0.195$). The proportion of cells constructed without alfalfa (using materials from the other plant species) was greater in the imidacloprid treatment (88.9%) than in the control (49.1%; odds ratio = 8.22, $p = 0.0346$; electronic supplementary material, figure S5), but was not correlated with irrigation treatment (odds ratio = 1.60, $p = 0.159$).

## 4. Discussion

Here, we demonstrate that application of a nursery formulation of imidacloprid to potted ornamental plants strongly affects the foraging activity and reproduction of a solitary bee. Plant irrigation level, in contrast, did not influence bee foraging or reproduction, but did affect the plants themselves. Lower irrigation resulted in higher imidacloprid concentrations in nectar of treated phacelia, and caused slight decreases in nectar volume and overall floral abundance in control cages. However, imidacloprid application seemingly negated the effects of irrigation on these floral metrics that we observed in untreated control plants. Despite affecting the plants directly, higher irrigation did not buffer bee foraging activity or

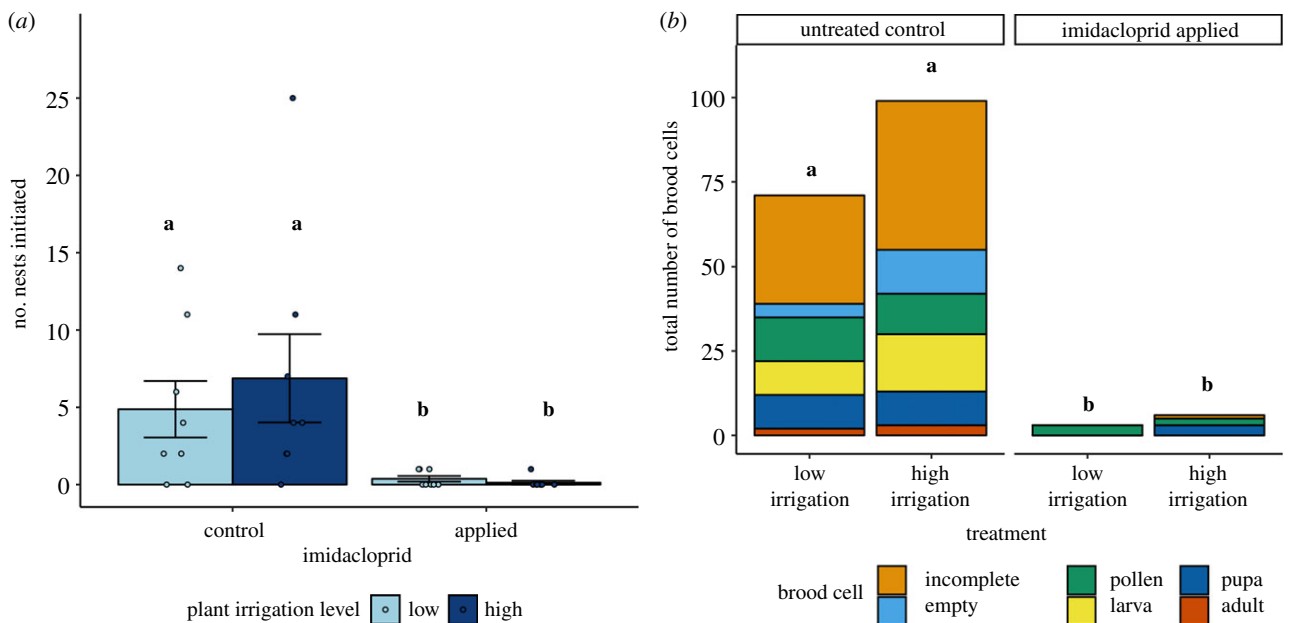

**Figure 3.** ALCB nesting responses to irrigation and imidacloprid treatments. (*a*) Nests initiated per cage. Each point represents one cage per study year ($N = 32$). (*b*) Cumulative number of cells constructed by treatment summed across cages and years. Bars not connected by the same lowercase letter were significantly different ($p < 0.05$) in post-hoc Tukey's HSD tests. (Online version in colour.)

reproduction against the negative impacts of imidacloprid application. Our study is, to our knowledge, the first to examine the consequences of a nursery neonicotinoid formulation on solitary bee reproduction. Our results have important implications for wild bee conservation in horticulture and other agroecosystems.

Applying a granular nursery formulation of imidacloprid at only 30% label rate reduced ALCB brood production by 90%. In general, granular formulations are understudied relative to seed and foliar applications [43]. Furthermore, field studies on neonicotinoids and ALCBs are surprisingly uncommon relative to laboratory exposure trials [44–47]. Field enclosure experiments with closely related solitary bees (*Osmia*) and neonicotinoid-treated crops, albeit using different application methods, found varying effects on bee reproduction [48,49]. Findings range from no observable effect in seed-treated *Brassica* [49,50] to a 50% decline in brood production with drench applications to wildflowers [12]. Open-field experiments on seed-treated *Brassica* yield even more disparate results, ranging from no effects [51–53] to a complete lack of nesting [48].

A potential reason for the stark reduction in reproduction we observed is the leaf clipping behaviour of nest building female ALCBs. Cutting leaves can result in contact exposure to systemic insecticides that would not be experienced by *Osmia* or eusocial bees [54–56]. We did not treat alfalfa in anticipation of it serving as the main nest building resource, yet 51.2% of all cells in our study contained no alfalfa. Instead, these cells comprised clippings of petals from *Baileya* and *Sphaeralcea* and leaves of the latter. While sample size was low, 8/9 cells collected from imidacloprid-treated cages were constructed exclusively with *Baileya* petals (electronic supplementary material, figure S5). While we did not analyse plant tissues for imidacloprid, studies report leaves containing higher neonicotinoid concentrations than nectar in treated plants [8,9,57] and high concentrations in whole-flower samples after label rate Marathon® application [14]. Thus, there are additional pathways by which ALCBs may encounter pesticides besides nectar and pollen consumption.

Exposure routes aside, we attribute the greater reductions in bee reproduction in our study relative to others in part to the insecticide's application method [58] and resulting concentrations in floral resources. Undoubtedly, this stems from pest management paradigms for nursery plants, where tolerance for aesthetic damage from pests is much lower than in field row crops [59]. In experiments with *Osmia*, neonicotinoid concentrations in seed-treated *Brassica* nectar are generally less than 15 ppb [48,49,51,53]. Based on phacelia nectar in our study, ALCBs encountered mean imidacloprid concentrations of at least 55 ppb, above the 'field realistic' range for seed-treated crop nectar [60,61] and rivalling 'maximum' concentrations for nectar in reviews [29,58,62]. Soil applications tend to result in higher neonicotinoid concentrations in floral resources compared to seed treatments, but the levels we documented exceed even the range of values reported for soil-treated crops [60]. Perhaps because plants in our study were confined to containers, imidacloprid did not leach as much from plants' root zones as it would in the ground. Our results emphasize the importance of formulation and dosage when assessing pesticide exposure risks to bees across crops and agricultural habitats.

We suspect the mechanism underlying the imidacloprid-associated reductions in ALCB reproduction was premature mortality in adults, rather than decreased nesting. This is supported by our occasional observations of dead bees on flowers only in imidacloprid cages. While oral $LD_{50}$ and $LC_{50}$ values of imidacloprid for ALCBs are not well established ([63], but see [64]), Cecala *et al.* [47] documented 29% and 68% reductions in adult ALCB longevity from ingesting 30 and 200 ppb sucrose syrups, respectively. In their experiment, survivorship fell to 50% after 6–12 days, depending on dosage. In ALCBs, most nesting activity occurs after the first week of adult emergence [30]. Although comparing effects between laboratory and field studies is difficult [65], other laboratory experiments [45,64] suggest imidacloprid is more toxic to ALCBs than to *Osmia*, and more detrimental than other compounds like insect growth regulators [64,66].

Furthermore, we suspect ALCBs were exposed to imidacloprid concentrations in nectar higher than those recorded in our accompanying phacelia nectar experiment. Samples from the nectar experiment were collected 15–18 weeks post-application (due to delays in onset of phacelia flowering), while ALCBs foraged on plants only 4–10 weeks post-application. Generally, neonicotinoid residues in plants decline over time, though this depends on numerous factors [16]. In 2019, six nectar samples taken from phacelia 5–8 weeks post-treatment with a 'high' dosage ranged from 63 to 219 ppb (low irrigation: $162 \pm 28$ ppb; high irrigation: $85 \pm 1.6$ ppb), but there were too few samples to permit further analysis. Applying a nursery formulation of imidacloprid to containerized plants, even at a reduced dosage well before bloom, may result in lethal concentrations in floral resources [16]. Our results support concerns about high concentrations and extended persistence of nursery formulations of neonicotinoids in plants [8–10,14].

Plant irrigation level did not affect ALCB reproduction. Even in untreated plants, we observed no benefits to bees from increased irrigation. Thus, we found no evidence of additive or synergistic effects between reduced irrigation and imidacloprid exposure. The lack of effect of irrigation could be due to our choice of native, drought-tolerant plants, though we had to ensure sufficient flowering even in our low irrigation treatment. Regardless, our low irrigation treatment is likely not as stressing as drought [22] or resource-limiting treatments in similar studies [12,36,67]. Rather, our low irrigation treatment mimicked a reduced watering regime, such as that which would be employed in a nursery, that avoids excessively water stressing plants. For example, Stuligross & Williams [12] found that imidacloprid exposure and resource limitation additively affected Osmia, though we are unable to compare pesticide exposure levels between studies as the authors reported no pesticide analyses of nectar or soil samples. Our results support the notion that reduced irrigation of potted ornamental plants, in the absence of pesticides, does not directly hinder solitary bee reproduction.

While we found no interactive effects between our imidacloprid and irrigation treatments on bees, we did detect interactions for containerized plants. Most interestingly, irrigation level mediated the effects of imidacloprid application in phacelia by affecting the amount of imidacloprid resulting in nectar, with higher concentrations in low irrigation plants. This pattern could be due to soil moisture and/or leaching rates [68]. First, low soil moisture causes plants to transpire more and increases xylem tension, resulting in higher water mobility and increased movement of water-soluble neonicotinoids [27,28]. Second, higher irrigation may lead to greater rates of imidacloprid leaching from potting soil [17,29]. While reduced irrigation may not diminish bee foraging or reproduction, it could result in elevated nectar concentrations of imidacloprid, which could indirectly harm bees. It remains to be seen if increased irrigation offsets the risks of neonicotinoid exposure at concentrations lower than those in this study. While neonicotinoid mobility in soil and plants in response to environmental conditions has received extensive study [29], we know of no research linking differing irrigation rates and soil moistures to neonicotinoid concentrations in floral resources. This topic deserves further investigation, particularly in bee-attractive plants [17].

Furthermore, application of our high imidacloprid dosage appeared to alter the effects of increased irrigation on floral resources that we observed in untreated plants. In untreated phacelia, higher irrigation positively influenced nectar volume. However, in phacelia treated with the high imidacloprid dosage, nectar volume did not differ with irrigation. We observed a similar trend for cage-level floral abundance in the ALCB reproduction experiment. High irrigation increased floral abundance, but only in untreated cages. In imidacloprid-treated cages, floral abundance did not differ with irrigation level. This pattern, in which imidacloprid alters the effects of reduced water on plants, could stem from the 'stress shield' phenomenon, whereby neonicotinoids purportedly offer plants resistance to abiotic stress by activating salicylate-associated defence pathways [69,70]. However, our results emphasize that any potential improvements to floral resources from neonicotinoid application do not compensate for the corresponding reproductive costs imposed on bees.

The effects of imidacloprid we document in this study may exceed those in a comparable field scenario. As in other field enclosure studies, bees were limited exclusively to flowering plants in cages, each of which (other than alfalfa) were treated with imidacloprid. In the field, a bee's foraging range might encompass plants both with and without pesticides. Available alternative forage can diminish the impacts of neonicotinoids on solitary bees [13]. However, previous work on wild bee foraging in nurseries suggests high day-to-day fidelity to floral patches [6]. The composition of wild Megachile pollen provisions also suggests a narrow use of available flowering species [71]. Therefore, it is reasonable that even patchily distributed pesticide-treated plants could result in chronic exposure for solitary bees. Field experiments explicitly examining solitary bee nesting in nurseries in relation to management practices are needed.

In conclusion, our results provide a broader understanding of how solitary bee reproduction can be impacted by local management practices in ecologically overlooked agricultural settings like nurseries. Specifically, we link the effects of local management practices to resources provided by flowering ornamental potted plants and solitary bee nesting and reproductive output, noting specific interactions which merit further study. Moreover, this work highlights important considerations for the conservation of wild bees in nursery systems.

Data accessibility. The datasets from this study are available from the Dryad Digital Repository: https://doi.org/10.6086/D1FX0D [72].

Authors' contributions. J.M.C.: conceptualization, data curation, formal analysis, funding acquisition, investigation, methodology, project administration, resources, software, validation, visualization, writing-original draft, writing-review, and editing; E.E.W.R.: conceptualization, funding acquisition, investigation, methodology, project administration, resources, software, supervision, visualization, writing-original draft, writing-review, and editing.

All authors gave final approval for publication and agreed to be held accountable for the work performed therein.

Competing interests. We declare we have no competing interests.

Funding. This work was supported by USDA NIFA Predoctoral Fellowship no. 2019-67011-29512 to J.M.C. and a California Association of Nurseries and Garden Centers grant.

Acknowledgements. We thank J.M.H. Jones, D. Baronia, K.J. Argumedo, T.H. Boyer, G.E. Lozano, D.T. Rankin, and UC Riverside Agricultural Operations for assistance with field work; Q.S. McFrederick for assistance with ALCBs; F.J. Byrne for assistance with ELISAs; and M.P. Daugherty for field cages. We acknowledge JWM Leafcutters Inc. for ALCBs and Moosa Creek Nursery for ornamental plants.

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
