## [Peer Review File · Proceedings of the Royal Society B: Biological Sciences]

Review History

RSPB-2021-0757.R0 (Original submission)

Review form: Reviewer 1

Recommendation

Accept with minor revision (please list in comments)

Scientific importance: Is the manuscript an original and important contribution to its field?

Excellent

General interest: Is the paper of sufficient general interest?

Good

Quality of the paper: Is the overall quality of the paper suitable?

Excellent

Is the length of the paper justified?

Yes

Should the paper be seen by a specialist statistical reviewer?

No

Do you have any concerns about statistical analyses in this paper? If so, please specify them explicitly in your report.

No

It is a condition of publication that authors make their supporting data, code and materials available - either as supplementary material or hosted in an external repository. Please rate, if applicable, the supporting data on the following criteria.

Is it accessible?

Yes

Is it clear?

No

Is it adequate?

Yes

Do you have any ethical concerns with this paper?

No

Comments to the Author

I really enjoyed this manuscript and very much appreciate its thoroughness. It is very well written. I have a few remarks and requests for further clarification in the methods. I think all the discussion and references cited are appropriate. Please see the attached remarks for details. (See Appendix A)

Review form: Reviewer 2

Recommendation

Major revision is needed (please make suggestions in comments)

Scientific importance: Is the manuscript an original and important contribution to its field?

Good

General interest: Is the paper of sufficient general interest?

Excellent

Quality of the paper: Is the overall quality of the paper suitable?

Excellent

Is the length of the paper justified?

Yes

Should the paper be seen by a specialist statistical reviewer?

No

Do you have any concerns about statistical analyses in this paper? If so, please specify them explicitly in your report.

No

It is a condition of publication that authors make their supporting data, code and materials available - either as supplementary material or hosted in an external repository. Please rate, if applicable, the supporting data on the following criteria.

Is it accessible?

Yes

Is it clear?

Yes

Is it adequate?

Yes

Do you have any ethical concerns with this paper?

No

Comments to the Author

This is an interesting and well-written study that was nicely prepared. The link with irrigation I found a little odd from a conceptual level and in the end there wasn't a whole lot going on with irrigation from a data standpoint (aside from some interactions for certain response variables), but the main effect of imidacloprid on a native bee alone is certainly worth studying and generated some very important data. It's wild that the label rate led to near 100% mortality (L108-110)! This is an important outcome, and even a 30% label rate reduced brood by 90%. This is incredible.

Even though water variation didn't generate much of an effect, I still think this aspect needs to be better justified. In particular, I think the authors need to better describe why they expect irrigation to interact with a soil-applied neonic. As is, they make a hypothesis and prediction at the end of the Introduction, but the potential interactions are never explicitly described or explored. This comes up a bit in the Discussion when explaining the data; however, the possible mechanisms need to be introduced upfront. These are water-soluble insecticides so more water could lead to more uptake, or it could lead to leaching from the pot and less uptake. Also, if more water leads to more flowers this could dilute the insecticide (i.e., same amount of compound split across more flowers), resulting in less concentrated neonics. These are a few possibilities (there are others), but none are introduced when explaining the rationale for why water and neonics might interact in a management context. This is a problem.

L25-27. Can you provide a better conclusion based on what you actually found? This closing sentences simply summarizes your general topic area but is not an outcome based on the data you collected.

L50. The connection between hedgerows and nurseries I found a little odd. Aren't hedgerows unique to agricultural fields?

L91 and 125-126. Presumably these species are also commonly sold as ornamental plants in nurseries (since this is the framing of the study)?

L95-97. Can you clarify the number of plants per cage? It's 30 plants per each of 6 treatments, but that doesn't fit evenly into 4 cages (2 blocks x 2 cages per block). Unless you meant it was one cage in each of two blocks; the wording there was unclear.

L98-103. Most people will have no idea what these rates represent. Are these what growers use? The levels of these treatments (i.e., amount of water) needs to be justified.

L104-110. Weren't there two doses? This only describes a high dose rate.

L111. When were irrigation treatments started? Unclear the progression here from starting plants (I assume these were started in a greenhouse – not specified) and when treatments imposed.

L115-116. I don't know what this means - "Eight days during bloom". Does that mean 8 days after bloom began?

L141. What was the source of these bees?

L382-388. This closing paragraph is merely a restatement of your results. The end of the Discussion is where you should be ending broadly about conclusions and implications, not recapping the results that have already been stated a few times.

L425. This paper was published in Ecology Letters, not Science

The figures in general are very nicely prepared, but figure 2 is pretty awful from a visual standpoint. It's nearly impossible to make out patterns. The data points are too small and, unless there was a 3-way interaction with time, why are these data shown over all 6 weeks? Please try to provide a summary figure here that averages over time.

Decision letter (RSPB-2021-0757.R0)

10-May-2021

Dear Mr Cecala:

I am writing to inform you that your manuscript RSPB-2021-0757 entitled "Pollinators and plant nurseries: how irrigation and pesticide treatment of native ornamental plants impact solitary bees" has, in its current form, been rejected for publication in Proceedings B.

This action has been taken on the advice of referees, who have recommended that substantial revisions are necessary. With this in mind we would be happy to consider a resubmission, provided the comments of the referees are fully addressed. However please note that this is not a provisional acceptance.

Sincerely,
Professor Gary Carvalho
mailto:proceedingsb@royalsociety.org

Associate Editor

Comments to Author:

Thank you for submitting this interesting manuscript, which has now been seen by two reviewers. Both were positive about your study, as am I, but both raised issues that need to be addressed. In particular, in your revision, please take care to address Reviewer 2's comment regarding provision of background and justification for your hypothesis. In the introduction, I also feel that more background to the potential importance of horticultural nurseries as "habitat" is necessary, given the low proportion of land-use area that such facilities cover. Finally, a smaller point: in your abstract, please take care to state the directionality of results (e.g. "modulates" is not clear).

Reviewer(s)' Comments to Author:

Referee: 1

Comments to the Author(s)

I really enjoyed this manuscript and very much appreciate its thoroughness. It is very well written. I have a few remarks and requests for further clarification in the methods. I think all the discussion and references cited are appropriate. Please see the attached remarks for details.

Referee: 2

Comments to the Author(s)

This is an interesting and well-written study that was nicely prepared. The link with irrigation I found a little odd from a conceptual level and in the end there wasn't a whole lot going on with irrigation from a data standpoint (aside from some interactions for certain response variables), but the main effect of imidacloprid on a native bee alone is certainly worth studying and generated some very important data. It's wild that the label rate led to near 100% mortality (L108-110)! This is an important outcome, and even a 30% label rate reduced brood by 90%. This is incredible.

Even though water variation didn't generate much of an effect, I still think this aspect needs to be better justified. In particular, I think the authors need to better describe why they expect irrigation to interact with a soil-applied neonic. As is, they make a hypothesis and prediction at the end of the Introduction, but the potential interactions are never explicitly described or explored. This comes up a bit in the Discussion when explaining the data; however, the possible mechanisms need to be introduced upfront. These are water-soluble insecticides so more water could lead to more uptake, or it could lead to leaching from the pot and less uptake. Also, if more water leads to more flowers this could dilute the insecticide (i.e., same amount of compound split across more flowers), resulting in less concentrated neonics. These are a few possibilities (there are others), but none are introduced when explaining the rationale for why water and neonics might interact in a management context. This is a problem.

L25-27. Can you provide a better conclusion based on what you actually found? This closing sentences simply summarizes your general topic area but is not an outcome based on the data you collected.

L50. The connection between hedgerows and nurseries I found a little odd. Aren't hedgerows unique to agricultural fields?

L91 and 125-126. Presumably these species are also commonly sold as ornamental plants in nurseries (since this is the framing of the study)?

L95-97. Can you clarify the number of plants per cage? It's 30 plants per each of 6 treatments, but that doesn't fit evenly into 4 cages (2 blocks x 2 cages per block). Unless you meant it was one cage in each of two blocks; the wording there was unclear.

L98-103. Most people will have no idea what these rates represent. Are these what growers use? The levels of these treatments (i.e., amount of water) needs to be justified.

L104-110. Weren't there two doses? This only describes a high dose rate.

L111. When were irrigation treatments started? Unclear the progression here from starting plants (I assume these were started in a greenhouse - not specified) and when treatments imposed.

L115-116. I don't know what this means - "Eight days during bloom". Does that mean 8 days after bloom began?

L141. What was the source of these bees?

L382-388. This closing paragraph is merely a restatement of your results. The end of the Discussion is where you should be ending broadly about conclusions and implications, not recapping the results that have already been stated a few times.

L425. This paper was published in Ecology Letters, not Science

The figures in general are very nicely prepared, but figure 2 is pretty awful from a visual standpoint. It's nearly impossible to make out patterns. The data points are too small and, unless there was a 3-way interaction with time, why are these data shown over all 6 weeks? Please try to provide a summary figure here that averages over time.

Author's Response to Decision Letter for (RSPB-2021-0757.R0)

See Appendix B.

RSPB-2021-1287.R0

Review form: Reviewer 2

Recommendation

Accept as is

Scientific importance: Is the manuscript an original and important contribution to its field?

Excellent

General interest: Is the paper of sufficient general interest?

Excellent

Quality of the paper: Is the overall quality of the paper suitable?

Excellent

Is the length of the paper justified?

Yes

Should the paper be seen by a specialist statistical reviewer?

No

Do you have any concerns about statistical analyses in this paper? If so, please specify them explicitly in your report.

No

It is a condition of publication that authors make their supporting data, code and materials available - either as supplementary material or hosted in an external repository. Please rate, if applicable, the supporting data on the following criteria.

Is it accessible?

Yes

Is it clear?

Yes

Is it adequate?

Yes

Do you have any ethical concerns with this paper?

No

Comments to the Author

I have no further suggestions. The authors did a great job on the revision and this revised paper is a very nice contribution. Good work!

Decision letter (RSPB-2021-1287.R0)

02-Jul-2021

Dear Mr Cecala

I am pleased to inform you that your Review manuscript RSPB-2021-1287 entitled "Pollinators and plant nurseries: how irrigation and pesticide treatment of native ornamental plants impact solitary bees" has been accepted for publication in Proceedings B.

The referee(s) do not recommend any further changes. Therefore, please proof-read your manuscript carefully and upload your final files for publication. Because the schedule for publication is very tight, it is a condition of publication that you submit the revised version of your manuscript within 7 days. If you do not think you will be able to meet this date please let me know immediately.

To upload your manuscript, log into <http://mc.manuscriptcentral.com/prsb> and enter your Author Centre, where you will find your manuscript title listed under "Manuscripts with Decisions." Under "Actions," click on "Create a Revision." Your manuscript number has been appended to denote a revision.

You will be unable to make your revisions on the originally submitted version of the manuscript. Instead, upload a new version through your Author Centre.

- 1) A text file of the manuscript (doc, txt, rtf or tex), including the references, tables (including captions) and figure captions. Please remove any tracked changes from the text before submission. PDF files are not an accepted format for the "Main Document".
- 2) A separate electronic file of each figure (tiff, EPS or print-quality PDF preferred). The format should be produced directly from original creation package, or original software format. Please note that PowerPoint files are not accepted.

3) Electronic supplementary material: this should be contained in a separate file from the main text and the file name should contain the author's name and journal name, e.g. `authorname_procb_ESM_figures.pdf`

All supplementary materials accompanying an accepted article will be treated as in their final form. They will be published alongside the paper on the journal website and posted on the online figshare repository. Files on figshare will be made available approximately one week before the accompanying article so that the supplementary material can be attributed a unique DOI. Please see: <https://royalsociety.org/journals/authors/author-guidelines/>

4) Data-Sharing and data citation

It is a condition of publication that data supporting your paper are made available. Data should be made available either in the electronic supplementary material or through an appropriate repository. Details of how to access data should be included in your paper. Please see <https://royalsociety.org/journals/ethics-policies/data-sharing-mining/> for more details.

<http://datadryad.org/submit?journalID=RSPB&manu=RSPB-2021-1287> which will take you to your unique entry in the Dryad repository.

Once again, thank you for submitting your manuscript to Proceedings B and I look forward to receiving your final version. If you have any questions at all, please do not hesitate to get in touch.

Sincerely,
Professor Gary Carvalho
<mailto:proceedingsb@royalsociety.org>

Reviewer(s)' Comments to Author:

Referee: 2

Comments to the Author(s).

I have no further suggestions. The authors did a great job on the revision and this revised paper is a very nice contribution. Good work!

Decision letter (RSPB-2021-1287.R1)

09-Jul-2021

Dear Dr Cecala

I am pleased to inform you that your manuscript entitled "Pollinators and plant nurseries: how irrigation and pesticide treatment of native ornamental plants impact solitary bees" has been accepted for publication in Proceedings B.

If you are likely to be away from e-mail contact please let us know. Due to rapid publication and an extremely tight schedule, if comments are not received, we may publish the paper as it stands. If you have any queries regarding the production of your final article or the publication date please contact procb_proofs@royalsociety.org

Data Accessibility section

Open Access

Paper charges

Sincerely,

Proceedings B

Appendix A

Line 18: Add authority and order: family for *Megachile rotundata* at first mention: *Megachile rotundata* F. (Hymenoptera: Megachilidae)

Line 70: As written, the subject (plural) and verb (singular) don't seem to agree. Rewrite: "The ALCB is a..."

Line 91: Cite Suppl Table 1 for showing the family name of this plant species and number of pots.

Line 102: Were all pots watered for the longer duration, even if all pots were not wilting and so that the high irrigation plants still received more water than the low ones?

Lines 112-115: Can you add in parentheses: 5 flowers X 6 days x 2 cages=reps to give the grand total of the points shown on Fig. 1a and b? I am not sure what the n-value are for samples to get nectar amount and sugar concentration.

Lines 116-117: Same as above for Fig. 1c. You measured 5 flowers per treatment on 8 days = 40 flowers x 2 cages, but then pooled nectar samples for measurements of imidacloprid. What is the n-value for the pooled samples (looks like 4/trt)?

Line 143: You should mention that you purchased overwintering prepupae from JWM Leafcutters in Canada to let reader know that you were working with commercially supplied bees, not wild ones. Also, you could mention the nursery when you site the plants.

Line 156: Did you weigh only straws with nests, and not the empty ones? Do you know when the nests were made? Cells made early in the season (or perhaps later in the summer in CA) will develop to adulthood in about 3 weeks, while those made later are more likely to be ones that enter diapause as prepupae. Do you think those destined to emerge as adults in the summer had ample time to emerge before putting them at 22C then 5C? Were there any bees that went to winter storage as pupae or teneral adults? These factors could help explain your lack of emergence after wintering. Furthermore, the provision masses may have been insufficient in quality or quantity to provide what was needed as nutrition to allow the bees to survive the winter. I am not sure ALCBs would naturally use asters or mallows since they prefer legumes (but definitely do ok on Phacelia). Lastly, my team has had overwintered bees from cage studies for which we cannot find a reason for low emergence, and other times emergence if great (without fluctuating temps).

Line 161: Please add detail in the description of incomplete vs empty cells and also of pollen (the label used in Fig. 3b). Does incomplete mean just leaf pieces in back or back plus edges? Does empty mean no provision, no egg on provision, or no cap on the end of the cell? Does pollen mean a provision mass (pollen plus nectar) without an egg or with a dead egg (or impression of an egg, which my team calls an egg scar)?

Line 174: You only seem to analyze the cage-level floral abundance. How do you think the abundance by species would have mattered for bee nesting? Did some plant species not offer much of a resource during your study period? Did you see bees foraging on all species? If not, how did that affect floral abundance for flower species actually utilized by the bees? You also noted that flower petals were used by the bees for nest construction (Supplmtl Fig. S4). ALCBs use flower petals even when nesting in alfalfa fields, so depending on what flowers were present (which may have not been equal in your cages), the

bees may have preferred the petals regardless of treatment. It would be helpful for the reader to know if the relative floral abundance by species was the same across cages and within treatments.

Line 180: When you refer to each straw as a replicate, it seems in this case to be all of them, even empty ones.

Line 183: What constitutes a cell? Sometimes the bees add leaf discs to the back of a straw and then never finish it or fill it with provision. Does your definition of cell mean that provision mass was made, which could be with or without an egg laid?

Lines 216-217: There is something weird with this sentence as written. Check for accuracy.

References: I think there are formatting errors and missing info for some of these.

Appendix B

10-May-2021

Dear Mr Cecala:

I am writing to inform you that your manuscript RSPB-2021-0757 entitled "Pollinators and plant nurseries: how irrigation and pesticide treatment of native ornamental plants impact solitary bees" has, in its current form, been rejected for publication in Proceedings B.

This action has been taken on the advice of referees, who have recommended that substantial revisions are necessary. With this in mind we would be happy to consider a resubmission, provided the comments of the referees are fully addressed. However please note that this is not a provisional acceptance.

Sincerely,

Professor Gary Carvalho
mailto: proceedingsb@royalsociety.org

Associate Editor

Comments to Author:

Thank you for submitting this interesting manuscript, which has now been seen by two reviewers. Both were positive about your study, as am I, but both raised issues that need to be addressed. In particular, in your revision, please take care to address Reviewer 2's comment regarding provision of background and justification for your hypothesis. In the introduction, I also feel that more background to the potential importance of horticultural nurseries as "habitat" is necessary, given the low proportion of land-use area that such facilities cover. Finally, a smaller point: in your abstract, please take care to state the directionality of results (e.g. "modulates" is not clear).

Authors' response:

We thank the Associate Editor and both reviewers for their insight and helpful feedback. We submit our point-by-point responses below and feel this has resulted in an improved manuscript.

In order to better establish background and justification for our hypothesis regarding interactions between irrigation and soil-applied neonicotinoids, we added a new paragraph to the Introduction (L. 77-88). See also response to Reviewer 2's comment below.

We provided additional rationale and expended upon the relevance of nurseries as habitat for bees (L. 43-46).

We edited our abstract to include the directionality of results (L. 23).

Reviewer(s)' Comments to Author:

Referee: 1

Comments to the Author(s)

I really enjoyed this manuscript and very much appreciate its thoroughness. It is very well written. I have a few remarks and requests for further clarification in the methods. I think all the discussion and references cited are appropriate. Please see the attached remarks for details.

Line 18: Add authority and order: family for *Megachile rotundata* at first mention: *Megachile rotundata* F. (Hymenoptera: Megachilidae)

Authors' response: We added the authority and higher taxonomy for *Megachile rotundata* (L. 19).

Line 70: As written, the subject (plural) and verb (singular) don't seem to agree. Rewrite: "The ALCB is a..."

Authors' response: We changed to "The ALCB is a [...]" (L. 93).

Line 91: Cite Suppl Table 1 for showing the family name of this plant species and number of pots.

Authors' response: We inserted the family name (Boraginaceae) in the text before the binomial species name (L. 118). Citing table S1 here, however, would be misleading, since this table and the number of pots in it correspond to the *ALCB reproduction* experiment, while it is the *nectar* experiment being discussed here. The sample sizes of phacelia plants in the nectar experiment is detailed on L. 122-125, and we edited this section for greater clarity in response to reviewer comments (L. 124-125). See also figure S1a.

Line 102: Were all pots watered for the longer duration, even if all pots were not wilting and so that the high irrigation plants still received more water than the low ones?

Authors' response: Yes. We clarified this (L. 131-132).

Lines 112-115: Can you add in parentheses: 5 flowers X 6 days x 2 cages= reps to give the grand total of the points shown on Fig. 1a and b? I am not sure what the n-value are for samples to get nectar amount and sugar concentration.

Authors' response: We added the total sample sizes, N , for the nectar volume and nectar sugar concentration models, respectively (L. 149-152). We also clarified how we reached these totals in the text.

Lines 116-117: Same as above for Fig. 1c. You measured 5 flowers per treatment on 8 days = 40 flowers x 2 cages, but then pooled nectar samples for measurements of imidacloprid. What is the n-value for the pooled samples (looks like 4/trt)?

Authors' response: We added the total sample size for the imidacloprid concentration model with clarification (L. 153-154).

Line 143: You should mention that you purchased overwintering prepupae from JWM Leafcutters in Canada to let reader know that you were working with commercially supplied bees, not wild ones. Also, you could mention the nursery when you site the plants.

Authors' response: We now state the respective origins of the bees (L. 189-190) and plants (L. 172) with additional specifics in the Acknowledgments.

Line 156: Did you weigh only straws with nests, and not the empty ones? Do you know when the nests were made? Cells made early in the season (or perhaps later in the summer in CA) will develop to adulthood in about 3 weeks, while those made later are more likely to be ones that enter diapause as prepupae. Do you think those destined to emerge as adults in the summer had ample time to emerge before putting them at 22C then 5C? Were there any bees that went to winter storage as pupae or teneral adults? These factors could help explain your lack of emergence after wintering. Furthermore, the provision masses may have been insufficient in quality or quantity to provide what was needed as nutrition to allow the bees to survive the winter. I am not sure ALCBs would naturally use asters or mallows since they prefer legumes (but definitely do ok on Phacelia). Lastly, my team has had overwintered bees from cage studies for which we cannot find a reason for low emergence, and other times emergence if great (without fluctuating temps).

Authors' response: We weighed all straws with nests or *any* evidence of nest construction, but also weighed a subset of empty straws to determine the average mass of an empty straw.

We do not have fine-scale (day-to-day) data as to date of initiation for each nest. We would expect that any non-diapausing adults laid towards the beginning of the experiment (mid to late June) would have definitely had time to emerge before being put at 22 °C / 5 °C. However, during nest dissection, we found no evidence of same-summer emergence (e.g., empty cocoons, frass in empty cells, dead fully-developed adults in cells). It is possible some bees laid later in the experiment did not have time to emerge, if they were non-diapausing. This situation is difficult to judge as we know of no data on southern California daylengths on propensity to diapause in ALCB (e.g., Pitts-Singer 2020). We fully agree that day-to-day data on nest initiation would help narrow the list of possible reasons why we observed no adult emergence. We appreciate your point about the quality and quantity of provision. We did not find immediate evidence that floral or nesting resources were limiting for bees in cages as mean nest mass did not decline with the number of nests constructed per cage (L. 323-325). We have added a brief mention of these topics to the Methods (L. 208-210).

Pitts-Singer, T. L. 2020. Photoperiod Effect on *Megachile rotundata* (Hymenoptera: Megachilidae) Female Regarding Diapause Status of Progeny: The Importance of Data Scrutiny. *Environmental Entomology* 49:516-527.

Line 161: Please add detail in the description of incomplete vs empty cells and also of pollen (the label used in Fig. 3b). Does incomplete mean just leaf pieces in back or back plus edges? Does empty mean no provision, no egg on provision, or no cap on the end of the cell? Does pollen mean a provision mass (pollen plus nectar) without an egg or with a dead egg (or impression of an egg, which my team calls an egg scar)?

Authors' response: We revised this section to describe in more detail the six categories into which we classified cells (L. 210-213).

Line 174: You only seem to analyze the cage-level floral abundance. How do you think the abundance by species would have mattered for bee nesting? Did some plant species not offer much of a resource during your study period? Did you see bees foraging on all species? If not, how did that affect floral abundance for flower species actually utilized by the bees? You also noted that flower petals were used by the bees for nest construction (Supplmtl Fig. S4). ALCBs use flower petals even when nesting in alfalfa fields, so depending on what flowers were present (which may have not been equal in your cages), the bees may have preferred the petals regardless of treatment. It would be helpful for the reader to know if the relative floral abundance by species was the same across cages and within treatments.

Authors' response: This is a good point. We did observe bees foraging on all plant species, which we now note in the Results (L. 305). However, we only have floral abundance data *by species* for the second year of the study (2020), not the first (2019). We have added a supplemental figure (now figure S4), showing the average floral abundance index for each of the five plant species in cages, by treatment, for 2020. This figure shows that there was substantial variation in the response of floral abundance to our treatments across the five plant species. We added wording about this new figure in the Results (L. 287-288).

Concerning flower petal use in nests: a pattern in which *Baileya* floral abundance was higher in imidacloprid treatments would explain more frequent use of petals (this species was the predominant source of petals used in nest building). However, post-hoc tests on a model of species-level floral abundance suggest *Baileya* floral abundance did not differ overall between control and imidacloprid treatments ($t_{37} = 1.369$, $p = 0.1794$); the only plant species for which this was the case was *Phacelia*, which ALCBs did not use at all for cell construction ($t_{37} = 5.551$, $p < 0.0001$). Please also refer to the new supplemental figure S4 for details.

Line 180: When you refer to each straw as a replicate, it seems in this case to be all of them, even empty ones.

Authors' response: This is correct, but only for our binomial GLMM (logistic regression). Each straw in the experiment was treated as a replicate, and was scored for whether it showed any evidence of nest initiation (yes) or no evidence of nest initiation (no). We clarified this in the Methods (L. 238-239).

Line 183: What constitutes a cell? Sometimes the bees add leaf discs to the back of a straw and then never finish it or fill it with provision. Does your definition of cell mean that provision mass was made, which could be with or without an egg laid?

Authors' response: We used "cell" to indicate all leaf disc structures constructed by ALCBs, regardless of whether a provision mass was made — in other words, all categories in Fig. 3b. We realize this could be confusing to readers, who may interpret "cell" more narrowly, as you described. In instances where we refer to all of these categories *collectively*, we have clarified our terminology (e.g., L. 211-212, 241-242, 246-249, etc.).

Lines 216-217: There is something weird with this sentence as written. Check for accuracy.

Authors' response: We edited this sentence to improve clarity and confirmed accuracy (L. 276-279).

References: I think there are formatting errors and missing info for some of these.

Authors' response: We double-checked each reference and corrected errors.

Referee: 2

Comments to the Author(s)

This is an interesting and well-written study that was nicely prepared. The link with irrigation I found a little odd from a conceptual level and in the end there wasn't a whole lot going on with irrigation from a data standpoint (aside from some interactions for certain response variables), but the main effect of imidacloprid on a native bee alone is certainly worth studying and generated some very important data. It's wild that the label rate led to near 100% mortality (L108-110)! This is an important outcome, and even a 30% label rate reduced brood by 90%. This is incredible.

Even though water variation didn't generate much of an effect, I still think this aspect needs to be better justified. In particular, I think the authors need to better describe why they expect irrigation to interact with a soil-applied neonic. As is, they make a hypothesis and prediction at the end of the Introduction, but the potential interactions are never explicitly described or explored. This comes up a bit in the Discussion when explaining the data; however, the possible mechanisms need to be introduced upfront. These are water-soluble insecticides so more water could lead to more uptake, or it could lead to leaching from the pot and less uptake. Also, if more water leads to more flowers this could dilute the insecticide (i.e., same amount of compound split across more flowers), resulting in less concentrated neonics. These are a few possibilities (there are others), but none are introduced when explaining the rationale for why water and neonics might interact in a management context. This is a problem.

Authors' response: This raises a good point. In the Introduction, we have added a more detailed justification and explanation of the potential interactions between soil-applied imidacloprid and irrigation level (L. 77-88), so that these points are raised earlier on in the manuscript.

L25-27. Can you provide a better conclusion based on what you actually found? This closing sentences simply summarizes your general topic area but is not an outcome based on the data you collected.

Authors' response: We revised the abstract to emphasize our actual data and highlight conclusions from this study (L. 25-28).

L50. The connection between hedgerows and nurseries I found a little odd. Aren't hedgerows unique to agricultural fields?

Authors' response: We had originally mentioned hedgerows here as an example of one type of floral resource supplementation by growers in agricultural areas. However, we agree its use as an example here reads very specific. We removed mention of hedgerows and revised the sentence to discuss, in general terms, the management strategy of adding floral resources to support beneficial insects (L. 60-63).

L91 and 125-126. Presumably these species are also commonly sold as ornamental plants in nurseries (since this is the framing of the study)?

Authors' response: This is true, especially at native plant nurseries. We clarified this (L. 173).

L95-97. Can you clarify the number of plants per cage? It's 30 plants per each of 6 treatments, but that doesn't fit evenly into 4 cages (2 blocks x 2 cages per block). Unless you meant it was one cage in each of two blocks; the wording there was unclear.

Authors' response: We now explicitly state that there were 45 plants per cage (L. 124-125). See also supplemental figure S1 for a schematic diagram of an experimental block.

L98-103. Most people will have no idea what these rates represent. Are these what growers use? The levels of these treatments (i.e., amount of water) needs to be justified.

Authors' response: This is a good point. The rates are relevant for the size of pots and the relative water availability between treatments. These rates are the predetermined rates for the model of spray stake emitters used, which are the market leading brand for irrigation of containerized plants. Given the size of pots we used in our experiment (2-L or "6-inch"), the spray stake used for the low irrigation treatment (gray) delivers the lowest flow rate available. The stake used for the high irrigation treatment (brown) delivers the second lowest flow rate, and is the only type with the same flow pattern (90° spray). We now briefly mention this in the Methods (L. 127-129).

L104-110. Weren't there two doses? This only describes a high dose rate.

Authors' response: Correct. We added an in-text description of the "low" dose here for clarity (L. 144-145).

L111. When were irrigation treatments started? Unclear the progression here from starting plants (I assume these were started in a greenhouse – not specified) and when treatments imposed.

Authors' response: For the phacelia nectar experiment, irrigation treatments inside field cages began on 14 January 2020, or 77 days after sowing and 75 days before bloom began. We have added dates to clarify this timeline (L. 119, 126).

L115-116. I don't know what this means - "Eight days during bloom". Does that mean 8 days after bloom began?

Authors' response: We revised this paragraph for clarification; i.e. x separate days total during the bloom period (L. 146-147, 152).

L141. What was the source of these bees?

Authors' response: We clarified that these were commercial population of bees purchased from Canada via JWM Leafcutters (L. 189-190).

L382-388. This closing paragraph is merely a restatement of your results. The end of the

Discussion is where you should be ending broadly about conclusions and implications, not recapping the results that have already been stated a few times.

Authors' response: We revised along suggested lines and included a broader discussion of the implications of this work (L. 466-478).

L425. This paper was published in Ecology Letters, not Science

Authors' response: We corrected the journal for Klaus *et al.* 2021 (L. 528).

The figures in general are very nicely prepared, but figure 2 is pretty awful from a visual standpoint. It's nearly impossible to make out patterns. The data points are too small and, unless there was a 3-way interaction with time, why are these data shown over all 6 weeks? Please try to provide a summary figure here that averages over time.

Authors' response: We agree that the original figure 2 was overly busy. We now present a revised version that plots means \pm s.e. per week (by treatment) to improve visual clarity. Given our analyses and general interest in how these patterns change over time, we strongly feel that time should be represented in this revised figure. Both cage-level floral abundance and, to a greater extent, bee foraging activity, declined over time in cages. By displaying the mean index \pm s.e. for each week, readers can more easily see patterns in the data among the four treatments, as well as over time (averaged between years). We have revised the figure legend to reflect the new visualization (L. 713-716).